

# Insight into naturally-charged Highly Oxidized Molecules (HOMs) in the boreal forest

Federico Bianchi[1], Olga Garmash[1], Xucheng He[1], Yan Chao[1], Siddharth Iyer[2], Ida Rosendahl[3], Zhengning Xu[4], Matti P. Rissanen[1], Matthieu Riva[1], Risto Taipale[1], Nina Sarnela[1], Tuukka Petäjä[1], Douglas R. Worsnop[1,5], Markku Kulmala[1], Mikael Ehn[1], Heikki Junninen[1]

[1]Department of Physics, University of Helsinki, Helsinki, 00014, Finland

[2]Department of Chemistry, University of Helsinki, Helsinki, 00014, Finland

[3]Arctic Research Centre, Aarhus University, Aarhus, 8000, Denmark

[4]Joint International Research Laboratory of Atmospheric and Earth System Sciences, School of Atmospheric Sciences, Nanjing University, 210023, Nanjing

[5]Aerodyne Research, Inc., Billerica, MA 01821, USA

*Correspondence to*: Federico Bianchi (Federico.bianchi@helsinki.fi)

**Abstract.** In order to investigate the role of the naturally charged highly oxidised molecules (HOMs) in the boreal forest we have performed measurements to chemically characterize the composition of negatively charged ions. Additionally, we compared this information with the chemical composition of the neutral (HOMs) detected in the ambient air during the same period. The chemical composition of the ions was retrieved using an Atmospheric Pressure interface Time-Of-Flight mass spectrometer (APi-TOF) while the gas phase neutral molecules (mainly sulphuric acid and HOMs) were characterized using the same mass spectrometer coupled to a nitrate-based chemical ionization unit (CI-APi-TOF). Overall, we divided the identified HOMs in two classes, HOMs containing only carbon, hydrogen and oxygen and nitrogen-containing HOMs or organonitrates (ONs). During the day, among the ions, in addition to the well-known sulphuric acid clusters, we found a large number of HOMs clustered with the two most common inorganic acids, nitrate ($NO_3^-$) or bisulphate ($HSO_4^-$), the first one being predominant. During the night, the detected ions were very similar to the neutral compounds and were mainly composed of HOMs clustered with $NO_3^-$.

For the first time, we identified several clusters contain up to 40 carbon atoms clustered with $NO_3^-$. At this regard, we think that these naturally charged clusters are formed by up to 4 oxidized α-pinene units.

Finally, diurnal profiles of the negative ions were consistent with the neutral compounds revealing that ONs peak during the day while non-nitrate HOMs are more abundant at night-time. However, during the day, a big fraction of the negative charge is taken up by the sulphuric acid clusters causing differences between detected neutral and ion HOM/ON species. As a result, the total signal of the ionised organic compounds was much lower during day than during the night.





## 1 Introduction

Ions are present everywhere in the atmosphere. They arise from, for example, galactic cosmic rays (GCR) and/or radioactive decay from the soil (radon and gamma) (Harrison and Carslaw, 2003; Hirsikko et al., 2011). The initial ions have generally a very simple structure. In the upper atmosphere, the primary ions are $O^+$, $O_2^+$ and $NO^+$, while in the dense air they are $N_2^+$, $O_2^+$, $O_2^-$ and $O^-$ (Smith and Spanel, 1995). Collisions of these ions with various trace gases lead to charge transfer to compounds with higher charge affinity and formation of a large variety of cluster ions. Negative charge is preferably transferred to acidic compounds like nitric acid, sulphuric acid and few other strong acid (lowest proton affinity), while positive charge is carried by basic compounds such as ammonia and amines (highest proton affinity) (Smith and Spanel, 1995). The production rate of the ions can vary depending on the altitude, location and the time of the year. In the boreal forest, such as Hyytiälä located at 61N, early spring average production rate calculated based on external radiation and radon measurements is about 4.5 ion pairs $cm^{-3}s^{-1}$ (Laakso et al., 2004).

It is well known that natural ions are able to enhance the formation rate of new particles and the mechanism is known as ion-induced nucleation (Raes et al., 1986; Yu and Turco, 2001; Kirkby, 2007; Arnold, 2008; Hirsikko et al., 2011). Recent laboratory experiments performed in the European Centre for Nuclear Research (CERN, CLOUD experiment) have systematically explored the influence of ions on new particle formation (NPF) in several different chemical systems. The presence of ions strongly enhanced pure sulphuric acid nucleation (Kirkby et al., 2011; Duplissy et al., 2016) as well as sulphuric acid – ammonia nucleation (Kirkby et al., 2011; Bianchi et al., 2012; Schobesberger et al., 2015; Kurten et al., 2016), while showed little to no effect on the sulphuric acid – amine nucleation (Almeida et al., 2013; Kuerten et al., 2014; Bianchi et al., 2014). Recently, the results from the same experiment revealed that ions can strongly enhance also pure organic nucleation in absence of sulphuric acid (Kirkby et al., 2016). Although Bianchi and co-workers (2016) have observed that new particle formation in the free troposphere depends on the availability of highly oxidized organic species, they have seen only a weak ion enhancement. Several studies have demonstrated also that the compounds participating in this process, the so-called Highly Oxidised Molecules (HOMs), play often a central role in NPF events (Kulmala et al., 1998; Ehn et al., 2014; Krechmer et al., 2015; Ortega et al., 2016; Kirkby et al., 2016; Bianchi et al., 2016).

HOMs can exist in the atmosphere both as ionized (naturally clustered with nitrate ions) and neutral species (Ehn et al., 2012; Ehn et al., 2014; Bianchi et al., 2016). Main source of HOMs to the atmosphere is likely the oxidation of terpenes, which are biogenically-emitted volatile organic compounds (BVOCs). The most abundant HOM precursor in the boreal forest is α-pinene, originating primarily from coniferous trees (Ehn et al., 2014; Jokinen et al., 2015). Ehn et al. (2014) have highlighted the formation of HOMs as first-generation oxidation products from the oxidation of monoterpenes. The HOMs are produced through the formation of peroxy radicals ($RO_2$) and subsequent intramolecular hydrogen-shifts followed by rapid reactions with oxygen, also called "autoxidation" (Crounse et al., 2013; Rissanen et al., 2014; Jokinen et al., 2015; Berndt et al., 2016). They are expected to contain a wide range of chemical functional groups, including hydroxyl, hydroperoxides, carbonyls and epoxides. Consequently, some of the HOMs have very low vapour pressures allowing them to react and/or condense nearly irreversibly onto aerosol surfaces (Trostl et al., 2016).

In oxidation of monoterpenes, such as α-pinene, characteristic HOMs contain similar amount of carbon and hydrogen to the parent molecule (i.e. $C_{10}H_{16}$), while the oxygen amount varies and can be as high as 13 atoms (Ehn et al., 2012; Ehn et al., 2014; Trostl et al., 2016; Kirkby et al., 2016). Beside monomer HOMs ($C_{10}$), dimer HOMs with an approximate composition of either $C_{19}H_{28}O_x$ or $C_{20}H_{30}O_x$ were also observed in both gas and particulate phases (Ehn et al., 2014; Lopez-Hilfiker et al., 2014). Rissanen et al. (2014) and Ehn et al. (2014) proposed that the bimolecular reactions of two peroxy radicals ($RO_2 + RO_2$) is one of the likely chemical pathways leading to the formation of dimers in the gas phase.





Organonitrates (ONs) were also identified in both gas and particulate phases from the oxidation of biogenic
compounds in the presence of $NO_x$ (NO + $NO_2$) and $NO_3$ radical (Ehn et al., 2014; Lee et al., 2016; Yan et al.,
2016). In addition to carbon, hydrogen and oxygen, these molecules contain at least one nitrogen atom. Different
reaction pathways leading to ON formation have been proposed. Due to their low vapour pressures, ONs are also
expected to have a potential important role in the formation and growth of secondary organic aerosol (SOA)
(Farmer et al., 2010; Kiendler-Scharr et al., 2016; Lee et al., 2016; Ng et al., 2017). In addition, Kulmala et al.
(2013) have recently proposed that the organonitrate $C_{10}H_{15}O_5NO_3$ is important for NPF. On the other hand
Jokinen et al. (2017) have shown the clustering and ONs are connected to each other during solar eclipse.
Although recently it has been demonstrated that the ions as well as the HOMs are very important during NPF
processes, their role in the boreal forest is still somewhat unclear. It has been shown that ion nucleation can
contribute up to 10% of the total nucleation, however it is still not known what is the role of the different ion
families (Kulmala et al., 2013). While few previous studies have shown the presence of naturally charged HOMs,
a careful comparison between naturally charged clusters and neutral organic compounds has not been attempted
so far. The aim of this study was to investigate the composition and diurnal changes of ambient ions, focusing on
the one composed by HOMs and ONs, and comparing them with the neutral species observed in previous studies.
**2    Materials and Methods**
All the measurements presented in this study were performed at the Station for Measuring Ecosystem-Atmosphere
Relations (SMEARII) located at Hyytiälä Forestry Field Station in Southern Finland (Hari and Kulmala, 2005)
during spring 2013, covering April, May and June. The SMEARII station is located on a flat terrain covered by a
homogeneous Scots pine (*Pinus sylvestris*) forest which is representative of the boreal coniferous forest. Two
Atmospheric Pressure interface Time-of-Flight (APi-TOF) (Aerodyne Research Inc. & Tofwerk AG; (Junninen et
al., 2010)) mass spectrometers in Hyytiälä were deployed to obtain data on naturally charged negative ions and
neutral molecules. An APi-TOF consists of a time-of-flight mass spectrometer (TOF) coupled to an atmospheric
pressure interface unit (APi) that allows sampling directly from the ambient air. The instrument that is used for
detection of neutral molecules is further equipped with a chemical ionization (CI) inlet (Jokinen et al., 2012).
In the APi-TOF, the naturally charged ions are sampled directly from the air (Junninen et al., 2010), while in
CI-APi-TOF, the ions are artificially created in the ambient pressure chemical ionisation inlet that is placed in
front of APi-TOF. The CI-APi-TOF inlet deployed for this campaign utilized nitrate-ionisation method and was
designed initially to measure neutral sulphuric acid, and later on was shown to also detect neutral HOMs and ONs
(Jokinen et al., 2012; Ehn et al., 2014). Chemical ionization is achieved by exposing clean air (sheath flow)
containing nitric acid ($HNO_3$) to alpha radiation (10 MBq $^{241}$Am source) or X-rays, which produces nitrate ($NO_3^-$)
ions. $NO_3^-$ ions in the sheath flow are directed into the sample flow by an electric field where they ionize the
ambient molecules by clustering (e.g. selected organic compounds) or proton transfer (e.g. sulphuric and some
dicarboxylic acids). The ionized molecules are then guided through a critical orifice to the TOF mass analyser.
$NO_3^-$ clustering ionisation is very selective to highly functionalised organic compounds (the molecule should
have at least two hydroxy or hydroperoxy groups), which makes this method ideal for measurement of HOMs
(Hyttinen et al., 2015). High resolving power of TOF mass analyser makes it possible to identify the chemical
composition of the detected molecules. The mass spectrometry data were processed and analysed using the
MATLAB-implemented latest version (6.03) of tofTools developed by Junninen et al. (2010).
For the purposes of this study, we will refer to non-nitrogen containing organics as "HOM-monomers" ($C_{10}$
molecules) and "HOM-dimers" ($C_{20}$ molecules). We will use "ONs" to refer to the nitrogen-containing HOMs.



When discussing ions, we will refer as "HOM/ON-nitrate" and "HOM/ON-bisulphate" to clusters of HOMs/ONs
with $NO_3^-$ and $HSO_4^-$ ions, respectively.
**3    Results and Discussion**
The focus of this work is to investigate the chemical composition of the naturally charged ions formed in the
Boreal forest and compare it with the neutral compounds detected by the CI-APi-TOF. Ehn et al. (2010) have
previously reported that during the day the main peaks observed are cluster composed by sulphuric acid, whereas
during night the major identified ions are HOMs clustered with $NO_3^-$. Day and night UMR (unit mass resolution)
spectra, averaged throughout the campaign, were analysed and detailed high-resolution mass spectra analyses are
provided for a typical clear sky day (diurnal and nocturnal spectra).
Figure 1 presents the average mass spectra of 10 clear-sky days during April and May 2013 of the negative
ions (Panel A and C) and neutral molecules (Panel B and D). The daily spectra (Panel A and B) are an average of
all of the mass spectra recorded from 09:00 to 13:00 (local time), while during the night (Panel C and D) the mass
spectra cover the time range from 23:00 until 03:00. Peaks with an odd and even mass-to-charge ratio (*m/z*) are
coloured in blue and red, respectively as a first indication of the nitrogen-containing molecules. This is based on
the nitrogen rule, where a deprotonated molecule/cluster containing one (odd) nitrogen will have an even mass
($HOM·NO_3^-$ and/or $ON·HSO_4^-$). While a deprotonated molecule/cluster containing zero or two (even) nitrogen
will have an odd mass ($ON·NO_3^-$ and/or $HOM·HSO_4^-$). The rule is reversed for radical species that can be
detected by CI-APi-TOF.
As previously reported by Yan et al. (2016), the mass spectra for the neutral compounds are dominated by
the species with an odd mass number ($ON·NO_3^-$) during daytime while during the night-time peaks with an even
mass number ($HOM·NO_3^-$) are the most abundant. From Figure 1, we can see the same pattern for naturally
charged clusters, although with some differences. While for the neutral compounds there is a clear odd/even
pattern during the day, the negative ions show only a small predominance of the odd masses. The main reason is
that while in the ionization unit of the CI-APi-TOF the HOMs and the ONs are ionized almost exclusively by the
primary ion ($NO_3^-$), in the atmosphere, during the day, the HOMs and the ONs can be ionized by either $NO_3^-$ or
$HSO_4^-$ ions. As ONs-nitrate will appear at odd mass and ON-bisulphate will appear at even mass, the difference
between odd and even *m/z* is less pronounced.
During the night, the organic species are mainly charged by $NO_3^-$ ions since the photochemical production of
sulphuric acid, and therefore $HSO_4^-$ ions is inhibited. As a result, the composition of naturally charged ions are the
same as the neutral molecules measured by the CI-APi-TOF making the two spectra (Panel C and D) in Figure 1
more comparable. In both mass spectra, we can still observe few peaks at high intensity with an odd atomic
number. These peaks are not ONs but are radicals formed from the ozonolysis of monoterpenes (*m/z* 325 -
$C_{10}H_{15}O_8·NO_3^-$ and 357 - $C_{10}H_{15}O_{10}·NO_3^-$ (Ehn et al., 2014; Yan et al., 2016)) and are highlighted in Panel D.
Contrary to a one large group of ions/neutral compounds within *m/z* 250 to 500 observed during the day, during
night two groups of molecules were distinguished. The first one, from *m/z* 250 to 450, are all identified as HOMs
clustered with $NO_3^-$ ions containing 8-10 carbon atoms. The second group (from *m/z* 450 to 650) are also HOMs
clustered with $NO_3^-$ ions but comprised of a larger number of carbon atoms ($C_{16}$-$C_{20}$) and are assigned to
HOM-dimers. Several studies have now reported that such dimer compounds are formed from the ozonolysis of
α-pinene (Ehn et al., 2014; Trostl et al., 2016; Kirkby et al., 2016). However, in this case, there are still some peaks
at odd masses (i.e. *m/z* 555: $C_{10}H_{31}O_{10}NO_3$) that have been attributed to night time $NO_3$ chemistry (Yan et al.,
161 2016).





### 3.1 Detailed chemical composition


To get further chemical information and confirmed as well the previous analysis we investigated a specific day by
high-resolution peak fitting. Here we show mass defect plots (Schobesberger et al., 2013; Bianchi et al., 2014) of
the negative ions and neutral molecules during the night (23:00 – 03:00) and during the day (09:00 – 13:00) on
April 20th 2013. In Figure 2, the top two panels report the chemical composition during night-time while the
bottom panels present the composition during daytime. In all the four mass defect plots, the coloured filled dots
correspond to the identified group of compounds described in the legend and the unfilled dots represent the
unidentified species. The size of the dot is proportional to the ion signal intensity of the different compounds.
As expected and partially shown in previous studies (Ehn et al., 2012; Yan et al., 2016), the mass defect plots
reveal that the chemical composition of the negative ions and the neutral compounds is different between day and
night. As shown in the UMR analysis, the neutral molecules as well as the negative ions indicate that, during the
day, ONs are formed at higher rate than during the night (light blue dots) while, during the night, non-nitrate
HOMs have higher concentrations (green dots). Interestingly, larger concentration of organic compounds such as
HOM-dimers could also be observed during the night, which is consistent with a decrease of the NO concentration
and subsequent increase of self- and cross-reactions of $RO_2$ radicals. From figure 2, we can clearly see that during
the night negatively charged ions and neutral compounds have similar composition. Several studies have
discussed that most of the HOMs detected during the night are formed by the reaction of monoterpene (e.g.
α-pinene) with ozone (Schobesberger et al., 2013; Ehn et al., 2014; Tröstl et al., 2016; Kirkby et al., 2016). Some
of these studies have also shown that via the same reaction (i.e. α-pinene ozonolysis) it is possible to form clusters
that contain several $C_{10}$-monomers ($C_{20}$, $C_{30}$ and $C_{40}$). In addition, Yan et al. (2016) have also observed the
formation of the HOM dimers ($C_{19}$-$C_{20}$) during the night. Besides these oligomers, other ONs (blue dots) were
also observed during night. These ONs are likely formed from the $NO_3$-initated oxidation of monoterpene. It is
worth noting that such behaviour has been observed previously for the neutral clusters, where the ONs and HOMs
are more abundant during the day and night-time, respectively (Yan et al., 2016).
The naturally charged ions reveal, however, additional information, which are mainly due to the fact that the
APi-TOF can also observe HOMs clustered with $HSO_4^-$ ions. Since sulphuric acid (and consequently the
bisulphate ion) is produced mainly during daytime from the OH-initiated oxidation of $SO_2$, the differences in
composition of ambient ions and neutral species is larger during the day. For example, at noon, all the major
naturally charged ions are composed by $HSO_4^-$ (*m/z* 97), sulphuric acid dimer ($H_2SO_4HSO_4^-$; *m/z* 195) and trimer
(($H_2SO_4)_2HSO_4^-$; *m/z* 293), while the neutral sulphuric acid measured by the CI-APi-TOF is by far not the
dominant peak. This is due to the strong electron affinity of sulphuric acid. The other remarkable feature in the
negative ion spectra is the daytime band of peaks (unfilled circles in Figure 2C) that spreads on the mass defect
plot wider than night-time dimers (Figure 2A) and day-time neutral species (Figure 2D). As shown in Figure 2,
these peaks are still unidentified. We suggest that some of the peaks reported in Figure 2C are HOMs and ONs as
seen in the neutral mass spectra, but clustered with $HSO_4^-$ ions. This is consistent with the fact that a lot of peaks
have a near zero or negative mass defect. To highlight the presence of these different species in the APi-TOF
during the day, a reference line (violet) was added. This line represents the most oxidised HOMs/ONs detected in
the neutral mode (Figure 2B). These compounds are the one with the lowest mass defect. By definition, in the
CI-APi-TOF, all the peaks appeared above the line. During the night-time, the naturally charged ions are also all
above the line, because in this case the ions are mainly formed by HOMs cluster with $NO_3^-$ ions, that is the same
mechanism inside the CI-APi-TOF. However, during daytime, the behaviour is totally different. The band is much
broader and many new peaks are situated below this line, suggesting that HOMs and ONs are clustered with
$HSO_4^-$ ions (orange dots). In addition, formation of naturally charged ions containing sulphuric dimer or trimer as



a core ion might be expected and could explain the formation of some ions observed solely during the day,
especially the one at really low mass defect. It is worth noting that sulphuric acid – HOM clusters have been
demonstrated to participate in NPF (Schobesberger et al., 2013; Riccobono et al., 2014) and similarly might be
involved in such processes in the boreal forest. This is the first time that such clusters have been detected in the
ambient.
In addition to the mass defect plots presented in Figure 2, the chemical composition of the naturally charged
negative ions measured during several nights the year before (for this specific case it was recorded the 13[th] of
March 2012) is presented in Figure 3. It is important to mention that during that time the instrument was tuned to
detect ions at really high masses and could likely explain why such observation was not possible during the 2013
campaign (Figure 2). In addition to two bands of monomers and dimers observed in Figure 2, Figure 3 depicts the
existence of larger molecules, likely trimer and tetramer clusters (or oligomers). The first band is mainly
composed by HOMs with roughly 9-10 carbon atoms, the second band with HOMs having 19-20 carbons. In
general, the four bands show that these clusters can contain up to 40 carbon atoms. These larger molecules were
previously detected during pure biogenic NPF in the CLOUD chamber from the ozonolysis of α-pinene (Kirkby et
al., 2016). This is also the first time that such compounds are recorded in the ambient atmosphere. Further studies
will be designed to investigate the formation of such species and to identify their potential impact in NPF.
As mentioned previously, $HOM^{\cdot}NO_3^-$ and $HOM^{\cdot}HSO_4^-$ adducts were identified using an APi-TOF. In Figure
4, the most abundant HOMs and ONs detected during the day of the 20[th] of April 2013 (same as Figure 3 C) are
presented. 10 HOMs (left panel) and 9 ONs (right panel) were chosen for comparison. The bottom side of the bars
referred to the HOMs/ONs clustered with $NO_3^-$ while the top part represents the signal intensity of the same
compounds clustered with $HSO_4^-$. We found that almost all the HOMs/ONs cluster more with $NO_3^-$ ions and on
average 60 % of the total signal ($HOMs/ONs^{\cdot}NO_3^-$ + $HOMs/ONs^{\cdot}HSO_4^-$) of these 19 compounds are clustered
with the nitrate.
**3.2    Diurnal variation of ions**
It is important to point out that different parameters could significantly impact the abundance of the ions detected
in the atmosphere. For instance, the charging of the organic molecules will be strongly affected by their
concentration, their proton affinity and their ability of forming clusters with acids like sulphuric acid or nitric acid.
Indeed, a competition between the different compounds getting the charge will depend on their physical-chemical
properties. For example, as a strong electro-negative compound sulphuric acid produced during the day will take a
large fraction of the negative charges available in the atmosphere. Such process will result in a reduction of other
naturally charged ions, while the concentration of the corresponding neutral species remains less than the
concentrations of other compounds. This phenomenon has to be taken into account while interpreting the diurnal
variation of the ions, especially during daytime. A direct comparison of the neutral HOMs with the corresponding
naturally charged HOMs will help to distinguished such processes since the CI-APi-TOF will measure the diurnal
evolution of the compounds independently of the ion availability in the ambient air.
While the evolution of the neutral HOMs/ONs have been already carefully described by Yan et al. (2016),
the diurnal evolution of the naturally charged HOMs is describe in the present study. Figure 5 shows the diurnal
cycle of ions measured over 7 clear-sky days during spring 2013. As expected, the diurnal trend of sulphuric acid
cluster-ions in Figure 5 tracks the solar radiation since $H_2SO_4$, during the day, is mainly photochemically
produced from the OH oxidation of $SO_2$. Around noon, three peaks attributed to sulphuric acid monomer, dimer
and trimer contribute to 20% of the total signal, which is comprised of several hundreds of species. Therefore,
during this time, the charge available for other compounds will be less. The sulphate signal anti-correlates with the
negatively charged HOM monomers and HOM dimers (i.e. clustered with $NO_3^-$) where they peak during the



night, while their concentrations in the daytime remain small as discussed previously. This is not only because
sulphuric acid is taking up the large fraction of the charge available but also because these HOMs are mainly
produced from the pure ozonolysis of α-pinene. However, it is not the case for the ONs from the reaction of $RO_2$
radicals with NO, which are, as sulphuric acid, mainly formed during the daytime. During the day, ONs either
cluster with $NO_3^-$ or $HSO_4^-$ ions, and their concentrations increase when the solar radiations rise. However, they
peak early morning before noon, then the concentration of sulphuric acid steeply limits the charge availability.
When sulphuric acid decreases because of the reduced sunlight, a second peak of the naturally charged ONs
clustered with $NO_3^-$ could be observed as shown in Figure 5. The ONs clustered with $HSO_4^-$ ions are less affected
by the sulphuric acid evolution since they are actually charged by $HSO_4^-$ ions that are increasing during that time.
257        Overall, such diurnal variation of these atmospheric ions is obviously influenced by the abundance of both
various neutral molecules and the charge carriers, as well as the charging efficiency between them. The neutral
HOMs are more abundant during the night-time, increasing at around 4pm and decreasing at 4am, owing to the
diurnal cycle of $NO_x$ (Yan et al., 2016). During this period, nitrate is the major charge carrier due to the inefficient
production of sulphuric acid, as a result, the $HOM \cdot NO_3^-$ clusters reveal an almost identical diurnal pattern with the
neutral HOMs. In correspondence with the decrease of neutral HOMs, the neutral ONs start to increase at around
4am in the morning, when the sulphuric concentration remains low and nitrate ion are still dominating, so a
corresponding increase of the $ON \cdot NO_3^-$ clusters are observed. These $ON \cdot NO_3^-$ clusters reach their maximum at
about 6am and started to decrease coincidentally with an increase of $ON \cdot HSO_4^-$ clusters. We attribute this change
to the shift of charge carrier from nitrate to bisulphate when the concentration of sulphuric acid is high enough to
take most of the ions. When the concentration of sulphuric acid become even higher, it become more competitive
in taking all the ions, and more importantly, in clustering with $HSO_4^-$. This leads to the reduction of $ON \cdot HSO_4^-$
clusters and the increase of $(H_2SO_4)_{0-3}HSO_4^-$ clusters. Two important details should be noted here: 1) the
concentration of neutral ONs also increase during the day and is usually more than one order of magnitude higher
than that of sulphuric acid, so the shift from $ON \cdot HSO_4^-$ to $(H_2SO_4)_{0-3}HSO_4^-$ clusters should be explained by the
higher charging efficiency (or clustering probability) between sulphuric acid and $HSO_4^-$; 2) such higher charging
efficiency could be due to the appearance of the "stabilizer", such as $H_2O$, $NH_3$, and amines (e.g. Kirkby et al.,
2011, Almeida et al., 2013). The reverse change in ion composition from midday to midnight can be explained by
the same underlying reasons.
276        Figure 6 shows the comparison of different ion families based on the hourly average during 5 sunny days
within the campaign period. Panel A in figure 6 shows the positive correlation between sulphuric acid clusters and
organic molecules (HOM+ON) charged by bisulphate ion. This confirms our identification of $HOM/ON \cdot HSO_4^-$
compounds as both signals are connected to photochemical production of sulphuric acid during the day. Both of
these ion families peak during the day and are absent during the night, which is consistent with figure 5. On the
other hand, sulphuric acid clusters anti-correlate with HOM Dimers (figure 6, panel C). As discussed before, the
dimers are ions containing two oxidised monoterpene units that are more abundant during the night as the
termination reactions of $RO_2$ radicals with e.g. NO are less likely. As the dimers are not produced efficiently
during the day, they are primarily charged with $NO_3^-$. Among the ionised species, dimers correlate with HOM
monomers that are charged with nitrate (figure 6, panel D); therefore, their diurnal profile will follow the green
line in figure 5. In panel D, we can observe two groups of points separated by night (blue) and day (green and
yellow) as the processes that control the formation of HOM Dimers and $HOM \cdot NO_3^-$ are similar: during the day it is
the termination reaction of $RO_2$ with NO, while during the night it is the abundance of ozone and monoterpenes.
Finally, in panel B of figure 6, we can see that HOM and ON charged with nitrate show positive dependence
during the day, while there is no correlation during the night. The night scatter results from the different formation
pathways: oxidation of monoterpenes with $NO_3$ radical is responsible for ON production, while ozonolysis is



responsible for HOM. During the day, the correlation between $ON \cdot NO_3^-$ and $HOM \cdot NO_3^-$ is coincidental and is
driven by the shift in the charger availability from nitrate to bisulphate, as also seen in figure 5.

## 4    Conclusion

Negative ions from the Finnish Boreal forest have been studied over a period of two months (March-April 2013).
In order to determine the ion chemical composition, we used an APi-TOF. The results have also been compared
with the chemical composition of the neutral compounds detected by the CI-APi-TOF. As expected, we found that
during the day the most intense ions are composed by sulphuric acid clusters, but this correspond to only 3 ions out
of the several hundred that were identified. We found that all the other peaks are mainly composed by HOMs or by
ONs clustered with $NO_3^-$ ions. In addition to that, we also observed clusters potentially important for new particle
formation composed by HOMs/ONs and $HSO_4^-$ ions. During the night, sulphuric acid concentration is extremely
low, as a result, the sulphuric acid clusters disappear. Therefore, also HOMs clustered with $HSO_4^-$ are not present
anymore. This lead to the fact that during the night, almost all the ions are formed by HOMs clustered with $NO_3^-$
ions, also the ONs are less abundant because of the low NO concentration during night, however we still observe
few ONs that arise from the $NO_3$-initated oxidation of monoterpene.
Comparing the chemical composition and diurnal variation of the ions with the neutral compounds measured
by the CI-APi-TOF we found that the HOMs detected are practically identical. The night-time spectra from the
two instruments are very similar. However, during the day, the spectra are quite different. First, the sulphuric acid
clusters are the major peaks. Second, the HOMs and the ONs can be detected in two different way, either cluster
with $NO_3^-$ or with $HSO4^-$ ions. This is the first time that bisulphate-organic clusters have been observed during
the day. This behaviour is confirmed during all the sunny days that has been analysed. Future studies will focus
more on the clustering of the HOMs with $HSO_4^-$ ions and comparing them with the days where we observe NPF.
Finally, it is important to note that for the first time we observed pure organic clusters that contain up to 40 carbon
atoms (4 α-pinene unit).

**Acknowledgments**

Liine Heikkinen, Qiaozhi Zha and Clemence Rose are acknowledged for useful discussions. Alexey
Adamov, Alessandro Franchin, Jonathan Duplissy, Maija Kajos, Putian Zhou, Simon Schallhart, Daniela
Wimmer, Arnaud Praplan, Mikko Äijälä, Tuija Jokinen, Juha Kangasluoma, Katrianne Lehtipalo, Mikhail
Paramonov, Ditte Taipale, Ella-Maria Duplissy, Siegfried Schobesberger, and the personnel of the Hyytiälä
forestry field station are acknowledged for help during field measurements. This work was partially funded by
Swiss National Science Foundation (grant P2EZP2_168787), Academy of Finland (299574, Finnish centre of
excellence 1141135), and European Research Council (COALA, grant 638703).





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





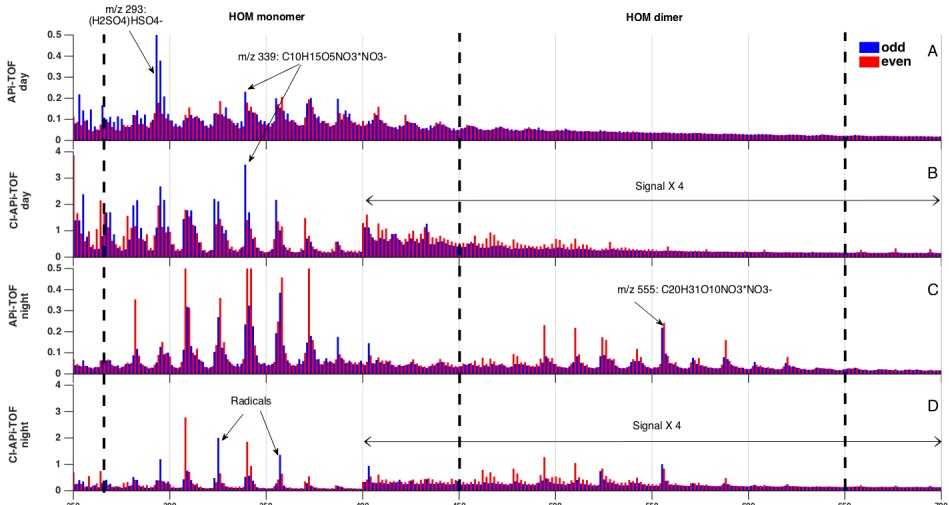

**Figure 1:** Averaged mass spectra of 10 days (clear sky condition) of measurements during April and May 2013.
The Y-Axes represent the peak intensity in counts s$^{-1}$. Panel A and B show, respectively, negative and neutral
clusters during the day (09:00-13:00). Panel C and D show, respectively, negative and neutral clusters during the
night (23:00-03:00). Odd masses have been colour coded in blue and even masses in red. The two black arrows in
panel B and D show the area of the spectrum where the signal have been multiplied by 4 (this was done only for
the CI-APi-TOF case).






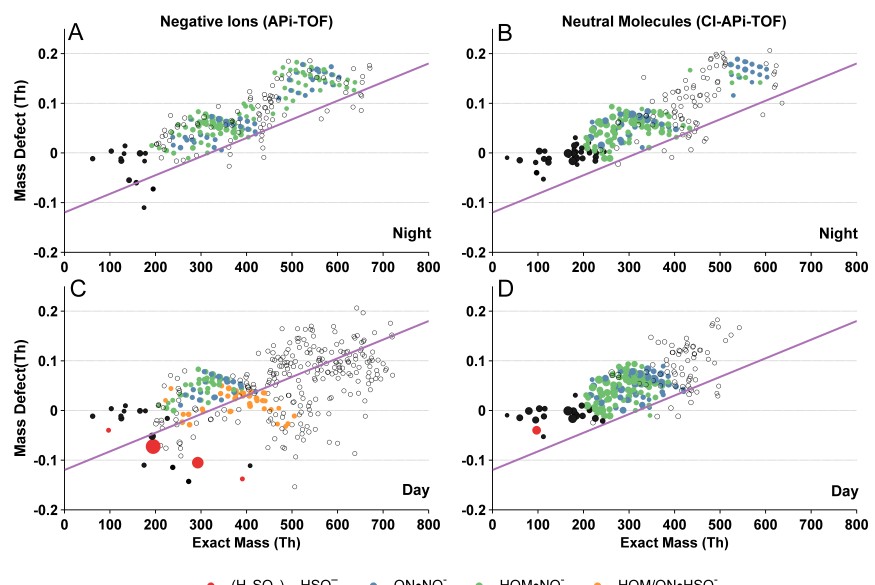


**Figure 2.** Mass defect plots for the neutral clusters and negative ions during the 20[th] of April 2013. Panel A and B
show the night time chemical composition of the negative and neutral clusters, respectively. Panel C and D show
the chemical composition during the day of the negative and neutral clusters, respectively. The size of the circle
represents the area of the peaks and is proportional to the detected amount. The compounds are coloured in
according to their chemical composition. Unfilled dots represent the unidentified compounds, while the black
filled dots represent other identified peaks as for example small organic acids. The violet line underlines the most
oxidised HOMs detected by CI-APi-TOF as clusters with $NO_3^-$ ions. Most probably most of the unidentified
negative ions that are placed below the line are HOM clusters with $HSO_4^-$ ions or $H_2SO_4HSO_4^-$ acid clusters.



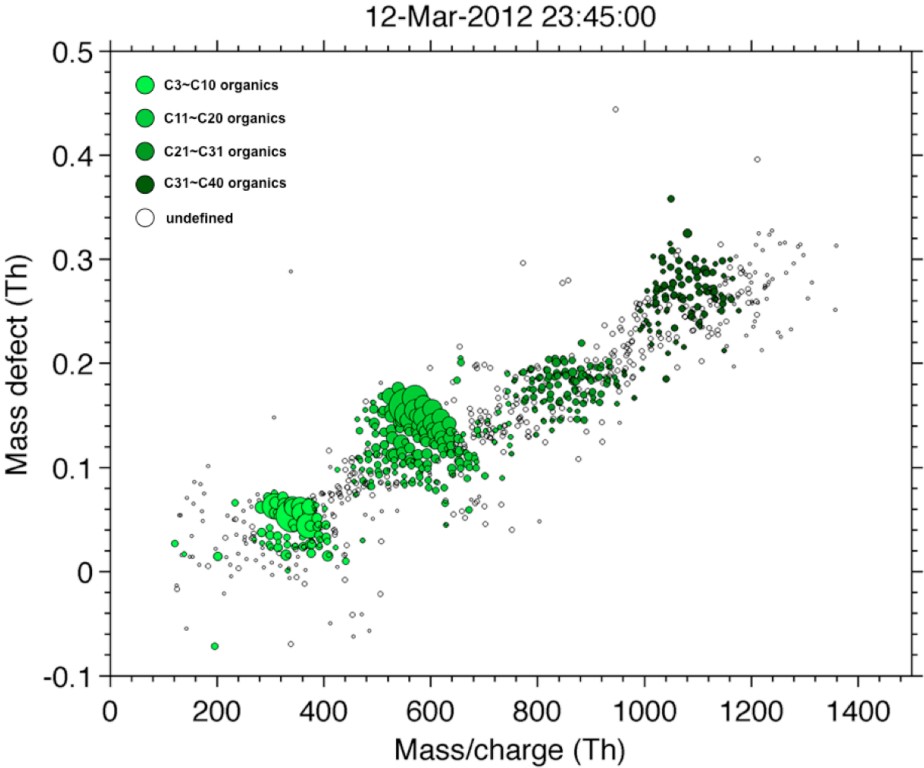


**Figure 3.** Mass defect plot of negative naturally charged ions observed during the night-time the 13[th] of March
2012. The four bands represent the HOMs containing approximately 10, 20, 30 and 40 carbon atoms (4 α-pinene
units). The majority of the HOMs have $NO_3^-$ as core ion.




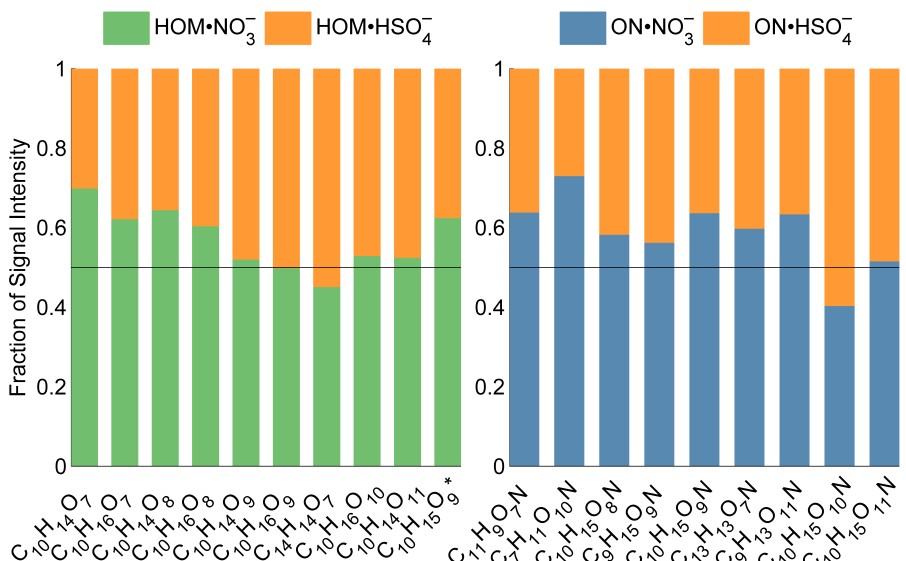


**Figure 4.** Naturally charged HOMs detected by the APi-TOF during daytime of April the 20[th]2013. On the left
panel, we report the naturally charged HOMs clustered ether with $NO_3^-$ (green) or $HSO_4^-$ (orange), while on the
right panel, we show naturally charged ONs clustered ether with $NO_3^-$ (blue) or $HSO_4^-$ (orange).




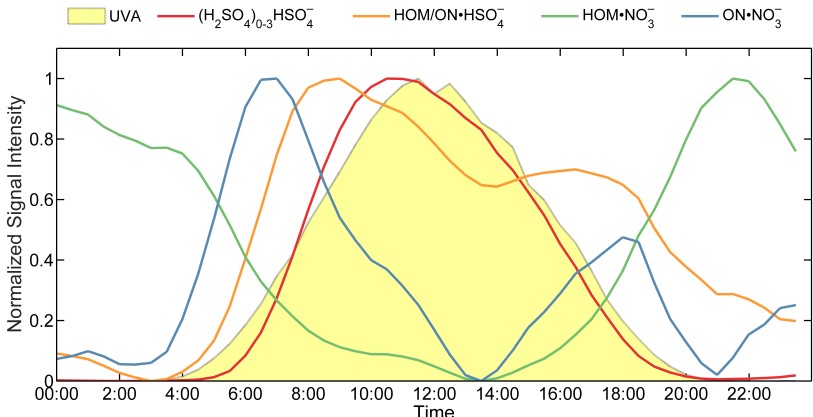


**Figure 5.** Averaged diurnal evolution of specific family compounds (ions) during days in clear sky conditions of
measurements done in April and May 2013. Colours and their corresponding families are denoted by the legends.
Each family is calculated by the sum of signals from the compounds of the family. The daily minimum of each
family is subtracted from the time trace, which is then normalized by the daily maximum. The HOM and ON
clustered with bisulphate ions (HOM/ON·HSO$_4^-$) have been grouped together since all these ions are present
mainly during the day.

589





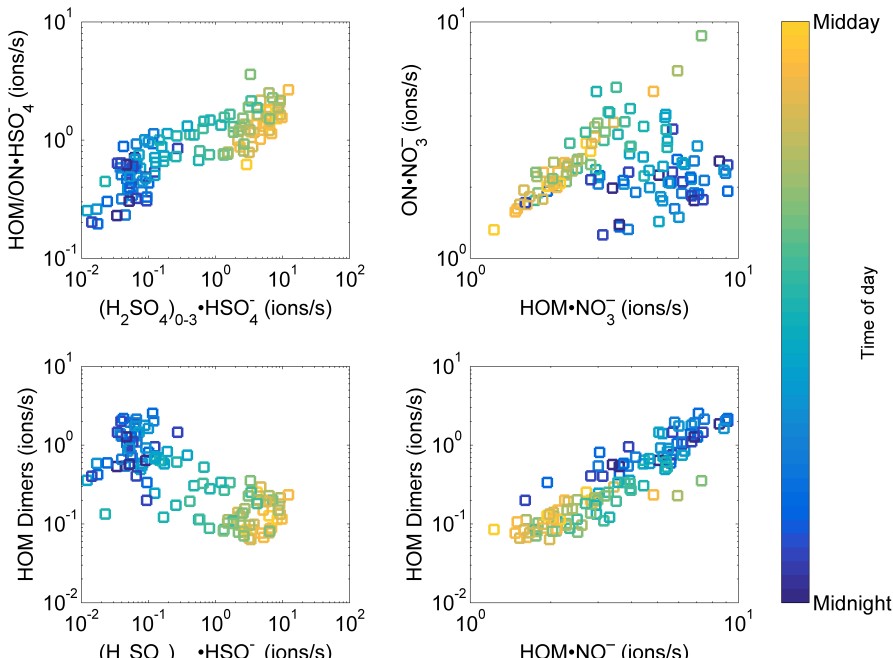

**Figure 6.** The diurnal dependency of naturally charged compounds detected by the APi-TOF. The data points are hourly averages for 5 sunny days during April-May 2013. The colour scale is normalised to show the period between midnight and midday, so that the transition between blue to green is around 6:00 and 18:00.