# Peer review of "The role of Highly Oxygenated Molecules (HOMs) and ambient ions characterized in the boreal forest."

_Atmospheric Chemistry and Physics, 2017_

## Referee Comment (RC1) · Anonymous Referee #1 · 29 Jun 2017

This manuscript presents measurements of naturally charged highly oxidized molecules (HOMs) by the APi-TOF and their neutral counterparts by the CI-TOF. HOMs were recently found to play important roles in new particle formation (NPF) and their exact formation mechanisms are still poorly understood. Ions on the other hand also play a role in particle formation especially in the free and upper troposphere. It is hence important to evaluate the roles of the charged and neutral HOMs in the process of forming nuclei. Since HOMs are primarily from oxidation of organic species in the atmosphere, considering the significant amount of organic species emitted to the boreal forest and the low abundance of sulfur species in the region, it is reasonable to relate particle formation with those HOMs. In addition, recent studies also pointed out

the potential dominant roles of HOMs in NPF in forested area such as Hyytiälä. The paper present interesting results and will potentially improve our understanding of NPF in forest where biogenic hydrocarbons are dominant VOCs in the air. The following issues need to be resolved before it goes to final publication.

1. The concentrations of naturally charged highly oxidized molecules (HOMs) are much lower (several orders, depending on the charging efficiency) than their corresponding neutral counterparts. Compared to the neutral HOMs, charged HOMs might exert little effects on NPF or if indeed they play important roles, the mechanisms will be likably very different from those of the neutral parts. Figure 1 shows the measured signals in cps which do not reflect the real concentrations. Did the authors convert those signals to concentrations i.e. number concentrations in cm-3? It is difficult to calibrate the instruments to give accurate concentrations but at least estimated concentrations can be obtained provided that a few assumptions were made. 2. The daytime and nighttime formation of organonitrates (ONs) was different, resulting probably in distinct chemical composition of the ONs between the two formation mechanisms. Can the authors provide further evidences of differences between the daytime and nighttime chemical compositions of the ONs? 3. According to the paper, there are at least four categories of HOMs: HOMs that contain only C,O,H or ONs that contain N in addition to C,O,H, and their corresponding NO3- or HSO4- clusters. The relationship between the naturally charged and neutral HOMs however has not been explored in details. Are there any correlations between them? i.g. the equilibrium or dynamic partitioning between them.

---

## Referee Comment (RC2) · Anonymous Referee #2 · 5 Jul 2017

Review of Bianchi et al. "Insight into naturally-charged [HOMs]..." This paper summarizes atmospheric observations of ambient ions from a forest site in Finland. This reviewer finds it to be of potential interest. Yet, while much of the observations have been attributed by the authors to have been presented before, poor presentation of the present results make discerning the new information difficult. This paper needs a 'sharp edge': a well-defined hypothesis and perhaps some sort of quantification of the reported species.

0. The authors may be puzzled by seemingly intentional misreadings but misnomers and poor phrasing have seriously hindered this reviewer's understanding of the authors'

intent.

1. Title " naturally-charged [HOMs]..." This wording suggests the core of the ions are HOM molecules, minus a proton. Or possibly they have large electron affinities, low ionization energies, or act as proton acceptors. The authors use of this terminology needs definition.

pts 2-7 in the abstract:

2. NO3- is not an inorganic acid. Furthermore, while HSO4- has a proton to donate, it is a very weak acid. These two ions are acting rather like bases in the atmosphere!

3. predominant aka most influential but here you specifically mean that nitrate has the highest abundance?
4. ' ions were very similar to the detected neutrals ' (the following phrase suggests the neutrals are actually ions, as does line 30.) You apparently mean that the masses of the HOM ligands on ambient NO3- ions are very similar to the masses of the HOM ligands on the NO3- produced in the CI machine? Or should we anticipate plots comparing abundances (relative or, best case scenario, absolute) ? See other loose terminology on this point (e.g. lines 172, 177.)
5. In the context of the preceding comment, the wording 'non-nitrate HOMs' is problematic (l 31).
6. Do these "several clusters..up to 40 C" comprise 4 separate 10 C molecular ligands ?
7. Line 34 suggests an important finding (or has it already been reported?) that HOMs and ONs do not cluster well with HSO4-. If the authors could provide semi-quantitative information on their relative ability to serve as molecular ligands to these two ions, that would provide a means to evaluate their (HOMs and ONs) roles in ion-induced NPF. Exploring this last point further:
l195-197 shows that HOMS and ONs do cluster with HSO4- but presumably weaker than they do with NO3-. Begs the question: How much weaker? Also, and this goes

to the choice of time periods (why is the sunlit data mostly before noon?) what is the mechanism for the evolution of the ions? Is it a quick evaporation of the HOMs/ONs (and an HNO3 ligand) once an HSO4- ion replaces the NO3- ion? Do the HOMs and ONs ligands get sequentially replaced by H2SO4 ligands ?

8. (also pts. 1 and 4) Using the word ionize in line 111 to describe what happens to HOM when it attaches to an ion is misleading. Please delineate whether you think HOM acquires a full e- of charge (we probably agree this is unlikely) and then put in your meaning of 'ionize'?

9. Paragraphs from line 124 to 161 reveal that much of what they observe has already been reported. Can any of these qualitative comparison be made quantitative in terms of abundances? Or perhaps there is an advance in this work over the previous ones where quantitative abundances of HOMs and ONs can be estimated ? Otherwise, there is a danger that nothing in this part is new...

9b. l 128/129 is confusing. Mentioned without much explanation are Unit mass resolution and high-resolution analyses: what are the and what are presented in the different figures ?

10. lines 144-147 states, rather pedantically, the fact that HSO4- hinders HOM and ON 'detection'. Strange to find it worded like that and also acknowledged so late, especially when this seems to be important for NPF. There seems to be a subtlety in the wording (146,147) that suggests ONs stick better to HSO4- than do HOMS ?

11. Figure 1 has large portions of the spectra that appear to be uninteresting. Could the information be better presented by focusing on certain sections of the mass spectra, say only 250-450 for this figure? Then a separate figure for the higer masses with a times 4 vertical axis for all four data sets... This reviewer is also interested in what ions are present below 250 amu. Can these be presented in an SI? The diurnal evolution of the bare NO3- ions (or are they clustered to HNO3 / H2O) would be interesting to see. In this vein, what fraction of ions have at their core, NO3- versus the total or vs. HSO4-? That information will provide for more points of discussion.

12. One side of Figure 2 is labeled 'neutral molecule' does this mean that the ionization

process has been identified (proton-transfer or the core ion and ligand, etc. ) such that the parent mass of the neutral species can be ascertained and then plotted ?

12b. To the untrained eye, these plots are massively defective in communicating quantitative information. It would help to have a legend showing circle size vs. ion intensity. It seems that most of the points are the same size, so only limited hope there. Would be extremely helpful for the uninitiated to have one or two of the most intense ions identified and their composition explained in detail, perhaps with a blowup of a select 'area' of data. Also, please identify the bare nitrate ions, which are apparently very low. The data in Fig. 4 is stated to be from Fig. 3c but this is probably Fig. 2C. These ions could be identified in some way (scoring/arrows?) in Fig. 2C. It is difficult to follow the discussion of the violet lines in these figures (lines 196-201). Finally, a succinct description and definition of 'mass defect' would be appreciated.

13. Fig. 4 needs a relative intensity indicator. Perhaps replacing the 50 14. Lines 221-227. Information on the ability to cluster to HSO4- vs. NO3- should be discussed here by presenting also the fraction: NO3- to the sum of NO3- and HSO4- core ion signals.

15. Fig. 5: Please provide a reasoning for subtracting the daily minimums in Figure 5. This seem to over-exaggerate tendencies in the measurements. COuld you provide an alternate plot, perhaps in a supplement, of log(signal/TotalSignal) vs. time , that is each ion signal family normalized by the same total ion signal? This plot has the potential to be more informative in an overall sense. The ratio of nitrate to bisulfate core ion signals (see pt. 14) would be a nice plot to see here also.

16. Presumably you have H2SO4 concentrations from CI. Please provide diurnal plot.

Something to ponder: The point of CI is to provide a definite ion-molecule reaction (IMR) time so that neutral abundancies can be ascertained. If relative intensities of ions are not much different in API-TOF mass spectra and the CI-mass spectra, then it is reasonable to postulate that the HOMs and ONs in a particular family have the same ion-molecule rate coefficient (one could furthermore speculate that it is near the

collisional rate!) Looking at 2a and 2b, stipulating that this reader understands these plots, it seems that the both sets of ONs (250-300 and 500-600) have about the same signal intensities whether allowed a long time to cluster with NO3- (ambient) or just a fraction of a second (CI). Thus the heavier ions with larger ONs do not seem to grow in time more than do the lighter ON ions. But the HOMS behave a little differently, where the higher mass set is more intense in the ambient ion spectra than in the short IMR. Is this an indication of sequential addition to ions of HOM monomer units and something different for ONs?

---

## Referee Comment (RC3) · Anonymous Referee #2 · 8 Jul 2017

13. Fig. 4 needs a relative intensity indicator. Perhaps replacing the 50 % black line with white line and then use a black line as a 'bar' indicator for each ion, all normalized to the most intense ion signal. Now the horizontal axis is identified by the ligand molecule. But again, they are not to be considered charged, either naturally or un-naturally, so as to be detected as HOM- or HOM+ from an ion that is stripped of ligands. This comment harkens all the way back to pts 1, 4, 8, etc. Do you want to identify the descriptor 'naturally-charged' to mean those ligands detected by API-TOF and the 'neutrals' to mean those detected by the un-natural NO3- ions produced in the CI machine? Perhaps it would be better to switch terminology: use ambient ions for naturally-charged ions.

[Figure]

14. Lines 221-227. Information on the ability to cluster to HSO4- vs. NO3- should be discussed here by presenting also the fraction: NO3- to the sum of NO3- and HSO4-core ion signals.
* * *

---

## Author Comment (AC1) · 16 Aug 2017

This manuscript presents measurements of naturally charged highly oxidized molecules (HOMs) by the APi-TOF and their neutral counterparts by the CI-TOF. HOMs were recently found to play important roles in new particle formation (NPF) and their exact formation mechanisms are still poorly understood. Ions on the other hand also play a role in particle formation especially in the free and upper troposphere. It is hence important to evaluate the roles of the charged and neutral HOMs in the process of forming nuclei. Since HOMs are primarily from oxidation of organic species in the atmosphere, considering the significant amount of organic species emitted to the boreal forest and the low abundance of sulfur species in the region, it is reasonable to relate particle formation with those HOMs. In addition, recent studies also pointed out the potential dominant roles of HOMs in NPF in forested area such as Hyytiälä. The paper present interesting results and will potentially improve our understanding of NPF in forest where biogenic hydrocarbons are dominant VOCs in the air. The following issues need to be resolved before it goes to final publication.

We would like to thank the referee for taking the time to read and comment on this manuscript and for the referee's helpful and constructive comments. In green, we report our answer to the reviewer and in blue, the changes applied to the manuscript.

1. The concentrations of naturally charged highly oxidized molecules (HOMs) are much lower (several orders, depending on the charging efficiency) than their corresponding neutral counterparts. Compared to the neutral HOMs, charged HOMs might exert little effects on NPF or if indeed they play important roles, the mechanisms will be likely very different from those of the neutral parts.  Figure 1 shows the measured signals in cps which do not reflect the real concentrations.  Did the authors convert those signals to concentrations i.e. number concentrations in cm-3? It is difficult to calibrate the instruments to give accurate concentrations but at least estimated concentrations can be obtained provided that a few assumptions were made.

Since both reviewers have mentioned the concentration argument, we have decided to make a new figure 5 where we directly compare the concentrations of the ambient ions with the neutral compounds. As the reviewer mentioned, it is difficult to calibrate the mass spectrometer at all the masses since the transmission curve inside the time-of-flight is not constant. However, it is definitely possible to estimate the concentrations of the ions, at least the order of magnitude.
Instead of just adding an axe in figure one we made an entirely new figure (the new figure 5), where we report the concentrations of the ions for the different families.

We produced two new figures reporting the ion concentration (Figure 5a and Figure 6). We also now discuss in the text these values and compared them with the concentration of the neutral ions.

2. The daytime and nighttime formation of organonitrates (ONs) was different, resulting probably in distinct chemical composition of the ONs between the two formation mechanisms.  Can the authors provide further evidences of differences between the daytime and nighttime chemical compositions of the ONs?

The reviewer is right. It is indeed true that the NOx chemistry is very different between daytime and nighttime. During the day, the chemistry is mainly related to NO while during the night the main

nitrogen containing oxidant is NO3 radical. This different chemistry has already been extensively discussed in Yan et al., ACP (2016).

Although, the study of Yan et al., describes that already quite well we decided to add some sentences/comments in this manuscript as well. Here we report just few of them.

*… In Figure 1, the neutral molecules as well as the negative ions indicate that, during the day, ONs, formed by the reaction between monoterpene, ozone and NO, have higher concentration than during the night (light blue dots) while …*

*…Interestingly, larger concentration of organic compounds such as HOM-dimers could also be observed during the night, which is consistent with a decrease of the NO concentration and subsequent increase of self- and cross-reactions of RO2 radicals…*

*…These ONs are quite different from the one observed during the day because they are formed by a different reaction, where the oxidation of monoterpene is initiated by the NO3 radical…*

3. According to the paper, there are at least four categories of HOMs: HOMs that contain only C, O, H or ONs that contain N in addition to C, O, H, and their corresponding NO3- or HSO4- clusters. The relationship between the naturally charged and neutral HOMs however has not been explored in details. Are there any correlations between them? i.g. the equilibrium or dynamic partitioning between them.

Actually, only two HOMs categories are discussed in this manuscript, the HOMs that contain only C, O, H or ONs that contain N in addition to C, O, H. These HOMs can easily cluster with NO3- or HSO4-. However, it is not by clustering with these deprotonated acids that the HOMs are different. We made sure that this point is now clear in the manuscript. Moreover, as mentioned in point 1, now we have added a new figure where we directly compare the concentrations of the ambient ions with the neutral species. Additionally, we have also added a figure (New figure 6) where we directly compare the ambient ions composed purely by pure sulphuric acid with the neutral sulphuric acid measured by the CI-APi-TOF.

Regarding the HOMs categories, in the abstract we state that:" Overall, we divided the identified HOMs into two classes: HOMs containing only carbon, hydrogen and oxygen and nitrogen-containing HOMs or organonitrates (ONs)."

As already mentioned it, we added two new figures (Figure 5A and Figure 6) where we directly compare ambient ions with neutral compounds.

---

## Author Comment (AC2) · 16 Aug 2017

Review of Bianchi et al. "Insight into naturally-charged [HOMs]..." This paper summarizes atmospheric observations of ambient ions from a forest site in Finland. This reviewer finds it to be of potential interest. Yet, while much of the observations have been attributed by the authors to have been presented before, poor presentation of the present results make discerning the new information difficult. This paper needs a 'sharp edge': a well-defined hypothesis and perhaps some sort of quantification of the reported species.

We would like to thank the referee for taking the time to read and comment on this manuscript and for the referee's helpful and constructive comments. In green, we report our answer to the reviewer and in blue, the changes applied to the manuscript.

0. The authors may be puzzled by seemingly intentional misreadings but misnomers and poor phrasing have seriously hindered this reviewer's understanding of the authors' intent.

We actually agree with the referee and in this revised version we tried to simplify the terminology and we modified the manuscript in order to reach a wider audience. We also agreed that our terminology is not entirely correct from a chemist point of view (eg. Naturally-charged HOMs).

In the new version, we also tried homogenise this problem as well. In the current version, we adopted the referee's terminology (e.g. ambient ions instead of naturally charged clusters).

1. Title " naturally-charged [HOMs]..." This wording suggests the core of the ions are HOM molecules, minus a proton. Or possibly they have large electron affinities, low ionization energies, or act as proton acceptors. The authors use of this terminology needs definition.

Although we can detect some HOM that have been ionised by deprotonation, this is definitely not the main ionization mechanism as pointed out by the referee in their review. We have now defined the terminology in the new manuscript where by naturally-charged HOM we mean an adduct/ligand formed by an HOM with a core ion that is usually a deprotonated strong acid as $NO_3^-$ and $HSO_4^-$. The title has been changed in order to avoid any confusion.

pts 2-7 in the abstract:

2. $NO_3^-$ is not an inorganic acid. Furthermore, while $HSO_4^-$ has a proton to donate, it is a very weak acid. These two ions are acting rather like bases in the atmosphere!

The referee is right. $NO_3^-$ and $HSO_4^-$ are the conjugated bases of the respective acids ($HNO_3$ and $H_2SO_4$). We have now referred to them as deprotonated inorganic acids.

3. predominant aka most influential but here you specifically mean that nitrate has the highest abundance?

We corrected our statement using the referee's suggestion.

4. 'ions were very similar to the detected neutrals' (the following phrase suggests the neutrals are actually ions, as does line 30.) You apparently mean that the masses of the HOM ligands on ambient NO3- ions are very similar to the masses of the HOM ligands on the NO3- produced in the CI machine? Or should we anticipate plots comparing abundancies (relative or, best case scenario, absolute)? See other loose terminology on this point (e.g. lines 172, 177.)

Here as well the referee is right with this sentence. We meant that the masses of the HOM ligands clustered to ambient NO3- ions are very similar to the masses of the HOM ligands clustered to the NO3- ions produced in the CI machine.
We changed the terminology here and in the manuscript especially in the suggested places. The whole manuscript has been corrected to avoid this confusion.

5. In the context of the preceding comment, the wording 'non-nitrate HOMs' is problematic (l 31).

We homogenise the terminology and we also correct this particular wording.

6. Do these "several clusters..up to 40 C" comprise 4 separate 10 C molecular ligands ?

Unfortunately, this is not known and our data set doesn't allow us to understand it any further. As the referee said, these compounds can either be formed by covalent bonds or ligands. In the new manuscript, we added a sentence to specify this point.

… However, while we know that the dimer is probably formed by a covalent bond between two α-pinene oxidized units, it is still not clear what is the bonding that formed these big clusters…

7. Line 34 suggests an important finding (or has it already been reported?) that HOMs and ONs do not cluster well with HSO4-. If the authors could provide semi-quantitative information on their relative ability to serve as molecular ligands to these two ions, that would provide a means to evaluate their (HOMs and ONs) roles in ion-induced NPF.
Exploring this last point further:
l195-197 shows that HOMS and ONs do cluster with HSO4- but presumably weaker than they do with NO3-. Begs the question: How much weaker? Also, and this goes to the choice of time periods (why is the sunlit data mostly before noon?) what is the mechanism for the evolution of the ions? Is it a quick evaporation of the HOMs/ONs (and an HNO3 ligand) once an HSO4- ion replaces the NO3- ion? Do the HOMs and ONs ligands get sequentially replaced by H2SO4 ligands?

We agree with referee that, how it is written now, Line 34 vaguely implies that HOMs and ONs do not cluster well with HSO4-. However, that was not the goal of our statement. Although this might be possible, this dataset can't provide the required semi-quantitative information. What we know is that (H2SO4)$_x$HSO4- clusters are (generally) stronger than HOM*HSO4- from the fact that there are more HOM than SA around.

However, currently there are no (even semi-quantitative) information on the relative efficiency of Org*NO3- vs Org*HSO4- clustering. This makes it impossible to infer about the likelihood of the different clusters - not to mention the dynamics of the clustering process, i.e. "the mechanism for the evolution of the ions". However, since the ions have a relative short lifetime, we can speculate

that the main mechanism is just a formation of an adduct between the deprotonated inorganic acid with a neutral HOM/ON without any displacement. But, so far, we have no proof of that.

In the new version of the manuscript we have rephrased that in order to avoid any confusion.

8. (also pts. 1 and 4) Using the word ionize in line 111 to describe what happens to HOM when it attaches to an ion is misleading. Please delineate whether you think HOM acquires a full e- of charge (we probably agree this is unlikely) and then put in your meaning of 'ionize'?

We have rephrased that sentence specifying the mechanism that lead to the detection inside the mass spectrometers of neutral gas phase molecules, whether they are HOMs or strong acid like sulphuric acid.

The new sentence is:
*… NO3– ions in the sheath flow are directed into the sample flow by an electric field where by forming an adduct (e.g. with HOM) or by proton transfer reaction (e.g. sulphuric and some dicarboxylic acids) neutral ambient molecules are charged and detected…*

In addition to that, and as mentioned already, the whole manuscript has been changed accordingly.

9. Paragraphs from line 124 to 161 reveal that much of what they observe has already been reported. Can any of this qualitative comparison be made quantitative in terms of abundances? Or perhaps there is an advance in this work over the previous ones where quantitative abundances of HOMs and ONs can be estimated? Otherwise, there is a danger that nothing in this part is new...
9b. line 128/129 is confusing. Mentioned without much explanation are Unit mass resolution and high-resolution analyses: what are the and what are presented in the different figures ?
10. lines 144-147 states, rather pedantically, the fact that HSO4- hinders HOM and ON 'detection'. Strange to find it worded like that and also acknowledged so late, especially when this seems to be important for NPF. There seems to be a subtlety in the wording (146,147) that suggests ONs stick better to HSO4- than do HOMS?

There are several new findings that we report in this paper that were probably not highlighted adequately. In this manuscript, we report for the first time the different clusters formed by HOMs and sulfuric acid during the day when nucleation is more important. In addition, we also report clusters containing ONs and sulphuric acid. Moreover, for the first time, we have drawn a detailed comparison between ambient ions and neutral species.

We have now rephrased the paragraph highlighting the new findings and tried to avoid all the confusion that the referee mentioned. For example, we did not intend to say that the ONs stick better to HSO4- than the HOMS. We believe that the higher concentrations of some of the specific ions are just due to the higher concentration of the respective neutral species. The reviewer has suggested a more quantitative comparison in a few of their comments. In response, we have prepared a new figure 5 where we directly report the concentrations of the ambient ions and neutral molecules. We believe that, by adding this extra figure, and by rephrasing the paragraph where we also address comments 9b and 10, this section is now more clear. The changes are highlighted in the manuscript.

11. Figure 1 has large portions of the spectra that appear to be uninteresting. Could the information be better presented by focusing on certain sections of the mass spectra, say only 250-450 for this figure? Then a separate figure for the higher masses with a times 4 vertical axis for all four data sets... This reviewer is also interested in what ions are present below 250 amu. Can these be presented in an SI? The diurnal evolution of the bare NO3- ions (or are they clustered to HNO3/H2O) would be interesting to see. In this vein, what fraction of ions have at their core, NO3- versus the total or vs. HSO4-? That information will provide for more points of discussion.

Although we understand why the reviewer would like us to divide the figure into two, we still prefer to have the HOMs region of the mass spectra all together. However, we have improved the figure, which now focuses on the region that the reviewer asked (250-650 m/z).
Regarding the ions below 250 amu, instead of adding additional SI material, we refer to the previous study made by Ehn et al., 2010. Figure 1 (taken from the mentioned study) shows the negative ion mass spectra for a typical daytime spectrum and a typical nighttime spectrum. It is possible to see from the figure that all the interesting ions below 250 amu have already been explained in detail in the previous work.

[Figure]

Figure 1. Negative ion mass spectra during the measurements in Hyytiälä, averaged over 3 h. (A) shows a typical daytime spectrum and (B) a typical nighttime spectrum. The ions are colored based on the identified composition. Figure taken from Ehn et al., 2010.

Figure 1 has been modified and in addition to that we have added the diurnal evolution of the bare NO3- ions in figure 5A. As already mentioned, the new figure 5A is now very useful to compare the different quantities.

12. One side of Figure 2 is labeled 'neutral molecule' does this mean that the ionization process has been identified (proton-transfer or the core ion and ligand, etc.) such that the parent mass of the neutral species can be ascertained and then plotted?

12b.   To the untrained eye, these plots are massively defective in communicating quantitative information. It would help to have a legend showing circle size vs. ion intensity. It seems that most of the points are the same size, so only limited hope there. Would be extremely helpful for the uninitiated to have one or two of the most intense ions identified and their composition explained in detail, perhaps with a blowup of a select 'area' of data.  Also, please identify the bare nitrate ions, which are apparently very low.  The data in Fig. 4 is stated to be from Fig.  3c but this is probably Fig.  2C. These ions could be identified in some way (scoring/arrows?)  in Fig.  2C. It is difficult to follow the discussion of the violet lines in these figures (lines 196-201).   Finally, a succint description and definition of 'mass defect' would be appreciated.

By neutral molecules, we refer to those compounds that were not charged prior to entering the ionisation region of our instrument (CI-APi-TOF). Therefore, it is true that what we show here is the adduct formed by the neutral molecule and the primary ion of our CIMS instrument ($NO_3^-$). We undoubtedly know the chemical compositions of the detected molecules, but we chose to still plot them in this way as it makes it easier to compare them with the ambient ions, where the same molecule goes through a similar mechanism but on a totally different time scale. To make this clear, our definition of neutral molecules has been elaborated in the text and in the caption.

A mass defect plot is a revealing way to present and compare mass spectra. In those plots, the abscissa represents the measured m/z of the compounds and the ordinate their mass defect, which is the difference between the accurate mass and the nominal mass (e.g., the exact mass of oxygen $^{16}O$ is 15.9949 Th and its mass defect is thus $-$ 0.0051 Th). In the new version, as suggested by the reviewer, the description of the plot has been improved. We also mention the fact that these plots are very powerful but are mainly useful for qualitative comparisons of mass spectra that include several hundreds of compounds. It is easier to see the difference in signal in Figure 1 or in the new figure 5a, where, following the referee's suggestion (points 15&16), we replotted the figure showing the neutral compounds as well. In addition, we don't use any normalisation anymore, but plot the concentration directly on a log scale so that all the time evolutions are visible.
We have also corrected the typo that referee mentioned (it is figure 2C and not figure 3C, as previously wrongly reported).

13. Fig. 4 needs a relative intensity indicator. Perhaps replacing the 50 % black line with white line and then use a black line as a 'bar' indicator for each ion, all normalized to the most intense ion signal.   Now the horizontal axis is identified by the ligand molecule.  But again, they are not to be considered charged, either naturally or un-naturally, so as to be detected as HOM- or HOM+ from an ion that is stripped of ligands.  This comment harkens all the way back to pts 1, 4, 8, etc.  Do you want to identify the descriptor 'naturally-charged' to mean those ligands detected by API-TOF and the 'neutrals' to mean those detected by the un-natural $NO_3^-$ ions produced in the CI machine? Perhaps it would be better to switch terminology:  use ambient ions for naturally-charged ions.

Regarding the suggested change for Figure 4 we don't think it is a good idea to add this extra line. Our scope it is just to compare how the different HOMs are clustering with the two different conjugated bases and not to show how intensive it is the signal of each HOMs. We think that keeping the figure simple will help the reader.

As previously mentioned, we agree with the reviewer regarding the terminology. As already mentioned we now have explained and simplified our terminology. The changes have been made consistently throughout the manuscript.

14. Lines 221-227. Information on the ability to cluster to HSO4- vs. NO3- should be discussed here by presenting also the fraction: NO3- to the sum of NO3- and HSO4- core ion signals.

We did not originally add that discussion because we thought it was too speculative. However, thanks to the reviewer's suggestion, we have now added a few lines about the core ion signal.
We detected a $NO_3^-$ signal of 0.065 cps (counts per second) and a $HSO_4^-$ signal of 0.034 cps. This means that $NO_3^-$ has a concentration that is almost factor of two higher than the bisulphate ion. This could explain why we see more clusters with $NO_3^-$ than with $HSO_4^-$. However, if we consider all the pure nitric acid clusters ($NO_3^- + HNO_3NO_3^-$) the signal is around (0.065+0.192) = 0.257 cps, while for all the pure sulphuric acid clusters ($HSO_4^- + H_2SO_4HSO_4^- + (H2SO_4)_2HSO_4^-$) the signal is around (0.035+6.7+3.4) = 10.135 cps. This means that sulphuric acid clusters have a concentration that is 40 times higher than that of the nitric acid clusters.

As requested by the reviewer this discussion has been added at the suggested place.

*…However, we should mention that we detected a $NO_3^-$ signal of 0.065 cps (counts per second) and a $HSO_4^-$ signal of 0.034 cps. This means that $NO_3^-$ has a concentration that is almost factor of two higher than the bisulphate ion. This could explain why we see more clusters with $NO_3^-$ than with $HSO_4^-$. However, if we consider all the pure nitric acid clusters ($NO_3^- + HNO_3NO_3^-$) the signal is around (0.065+0.192) = 0.257 cps, while for all the pure sulphuric acid clusters ($HSO_4^- + H_2SO_4HSO_4^- + (H_2SO_4)_2HSO_4^-$) the signal is around (0.035+6.7+3.4) = 10.135 cps. This means that sulphuric acid clusters have a concentration that is 40 times higher than that of the nitric acid clusters, showing once more that the sulphuric acid ions are the dominant peaks and that they cluster together very effectively…*

15. Fig. 5: Please provide a reasoning for subtracting the daily minimums in Figure 5. This seem to over-exaggerate tendencies in the measurements. COuld you provide an alternate plot, perhaps in a supplement, of log(signal/TotalSignal) vs. time, that is each ion signal family normalized by the same total ion signal? This plot has the potential to be more informative in an overall sense. The ratio of nitrate to bisulfate core ion signals (see pt. 14) would be a nice plot to see here also.

16. Presumably you have H2SO4 concentrations from CI. Please provide diurnal plot. Something to ponder: The point of CI is to provide a definite ion-molecule reaction (IMR) time so that neutral abundancies can be ascertained. If relative intensities of ions are not much different in API-TOF mass spectra and the CI-mass spectra, then it is reasonable to postulate that the HOMs and ONs in a particular family have the same ion-molecule rate coefficient (one could furthermore speculate that it is near the collisional rate!) Looking at 2a and 2b, stipulating that this reader understands these plots, it seems that the both sets of ONs (250-300 and 500-600) have about the same signal intensities whether allowed a long time to cluster with NO3- (ambient) or just a fraction of a second (CI). Thus the heavier ions with larger ONs do not seem to grow in time more than do the lighter ON ions. But the HOMS behave a little differently, where the higher mass set is more intense in the ambient ion spectra than in the short IMR. Is this an indication of sequential addition to ions of HOM monomer units and something different for ONs?

We combined point 15 and point 16 because they are strictly related.

Regarding the old figure 5, we agree with the reviewer that it over-exaggerates the tendencies in our measurements. Instead of providing a reason of our previous figure we decide to follow completely the reviewer suggestions and we therefore provided a new figure where we show the concentrations of the ions family. In the new figure, we also added the nitrate signal so that make it easier to compare the sulphate ions with the nitrate one. Additionally, we have also added a plot which includes the respective family in the neutral mode, that include sulphuric acid as well as requested.

As already discussed in a previous reviewer comment, we agree that the mass defect plot is not the best way to compare quantities. Figure 5 now shows concentrations instead of normalized signals. Comparing the two plots in figure 5, the difference in signals between the different modalities is clearly visible. Figure 2 has been kept as is as we still think that the mass defect plot is quite useful for a qualitative comparison.

These information are now presented in the new figure 5 (Panel A & B) where we report the variation of the neutral species during the day. As requested by the reviewer, we don't normalize the signal anymore and only report the concentration on a log scale. Obviously, the variation is less pronounced but it is still very visible. In addition, we have also added an extra figure where we compare the ambient ions purely formed by sulfuric acid and their clusters with the sulfuric acid concentration.